# Reproductive Microbiomes in Domestic Livestock: Insights Utilizing 16S rRNA Gene Amplicon Community Sequencing

**DOI:** 10.3390/ani13030485

**Published:** 2023-01-31

**Authors:** Rebecca K. Poole, Dallas R. Soffa, Brooke E. McAnally, Molly S. Smith, Kyle J. Hickman-Brown, Erin L. Stockland

**Affiliations:** Department of Animal Science, Texas A&M University, College Station, TX 77843-2471, USA

**Keywords:** livestock, microbiome, reproduction, sequencing

## Abstract

**Simple Summary:**

The reproductive microbiome includes bacterial colonies within the vagina, uterus, placental tissues and fluids, semen, and milk. Assessments of the human reproductive microbiome have established commensal and pathogenic bacteria that can influence fertility. Reproductive efficiency in domestic livestock species (dairy and beef cattle, swine, sheep and goats, and horses) is also impacted by bacterial presence and abundance within the reproductive tract of both males and females. However, advancements from culture-based methods of bacterial identification to specific sequencing of the 16S rRNA gene have only recently begun to establish this interconnected relationship. Thus, this review aims to discuss the most recent overview of reproductive microbial identification in domestic livestock species utilizing 16S rRNA gene sequencing and its potential roles in fertility.

**Abstract:**

Advancements in 16S rRNA gene amplicon community sequencing have vastly expanded our understanding of the reproductive microbiome and its role in fertility. In humans, *Lactobacillus* is the overwhelmingly dominant bacteria within reproductive tissues and is known to be commensal and an indicator of fertility in women and men. It is also known that *Lactobacillus* is not as largely abundant in the reproductive tissues of domestic livestock species. Thus, the objective of this review is to summarize the research to date on both female and male reproductive microbiomes in domestic livestock species (i.e., dairy cattle, beef cattle, swine, small ruminants, and horses). Having a comprehensive understanding of reproductive microbiota and its role in modulating physiological functions will aid in the development of management and therapeutic strategies to improve reproductive efficiency.

## 1. Introduction

Bacteria identification was previously evaluated using culture-dependent methods which limit the scope of species identification as only about 2% of bacteria can be cultured in a laboratory [1]. Recent advancements in culture-independent methods and 16S rRNA gene amplicon community sequencing have vastly expanded the understanding of the reproductive tract microbiome. The NIH Human Microbiome Project (HMP), launched in 2007, sought to understand the complexity and role of the microbiome [2]. In humans, reproductive microbiomes have been identified as an indicator of fertility. The genus *Lactobacillus* accounts for over 90% of the vaginal microbiome in healthy women, and a reduction in *Lactobacillus* dominance and greater bacterial diversity is associated with compromised fertility [3,4,5]. In men, bacteriospermia (>1000 colony-forming units/mL of semen) negatively impacts sperm motility, with *Lactobacillus* spp. being associated with improvements in sperm quality and *Prevotella* spp. having negative effects [6,7]. Following the HMP, a multitude of studies began profiling reproductive microbiomes in other mammalian species. This review summaries the research to date on reproductive microbiomes in domestic livestock.

## 2. 16S rRNA Gene Amplicon Community Sequencing

The 16S rRNA gene codes for the RNA component of the 30S subunit of the prokaryotic ribosome that is ubiquitous in bacteria and archaea, and this gene is comprised of approximately 1550 base pairs and hypervariable regions (V1 to V9) [8]. For 16S rRNA gene sequencing, one or more hypervariable regions are amplified and then sequenced, and the information gathered allows for the taxonomic composition and diversity [9]. This method provides information on which microbes are present and the relative abundance or percentage of a total population attributed to one taxon (i.e., phyla or genera) within a sample [9]. Additionally, bacterial diversity estimates and comparisons within and/or between samples can be measured. Alpha diversity evaluates the different numbers of bacterial species within a sample and can evaluate the richness and/or evenness of bacteria distribution [9]. The richness, represented as Chao1 metrics, refers to the number of different bacterial species detected, the evenness of distribution refers to how balanced the bacterial species are within a sample, and the Shannon diversity metric captures both the richness and evenness [9,10,11]. Beta diversity calculates the differences in diversity between samples and is presented as a distance or dissimilarity matrix such as Bray–Curtis or UniFrac [9,12,13]. While this sequencing technology has vastly expanded knowledge about microbial species and the microbiome, limitations do exist. It cannot account for the activity or metabolic potential of a microbial community, and it may not differentiate between individual bacterial species or strains (e.g., distinguish between pathogenic and commensal strains of *Escherichia coli*), determine if a microbe is alive or dead, or reflect a total microbial load [9]. Nevertheless, 16S rRNA gene amplicon community sequencing can provide helpful insights towards the understanding of the reproductive microbiome.

## 3. Dairy Cattle

### 3.1. Female Reproductive Microbiome

#### 3.1.1. Vaginal Microbiome

In humans, the presence of *Lactobacillus* spp. is commonly reported within the vaginal microbiome as important bacteria that could potentially aid in the establishment of pregnancy [3,4,5]. However, in dairy cattle, the genus *Lactobacillus* is not as commonly identified and typically detected at low abundance (<1%) [14]. The microbial population of the vagina in dairy cattle, because of its location in relation to the external environment, has been associated with increased levels of both bacterial diversity and richness compared to other sections of the reproductive tract, including the uterus [14]. Similar to the microbiomes of other reproductive organs, the most abundant phyla in the dairy cattle vagina are Firmicutes, Proteobacteria, Bacteroidetes, and Actinobacteria. Some studies have also noted the presence of Tenericutes and Fusobacteria as abundant phyla [15,16,17,18]. One study did note that as dairy cattle progress through a normal estrous cycle with changes in progesterone and estradiol concentrations, there are associated changes in the vaginal microbiome, including an increase in *Lactobacillus* abundance (with phylum Firmicutes) during the follicular phase, though the relative abundance of Lactobacillus was still low at 0.2% [18]. Identification through 16S rRNA sequencing has also established across multiple studies that one of the most prevalent genera is *Ureaplasma*. This has been repeatable across multiple studies, leading to the implications that *Ureaplasma*, in addition to its close relation with *Mycoplasma* (within the class Mollicutes and phylum Tenericutes), is associated with multiple reproductive issues in cattle. However, the establishment of *Ureaplasma* within the vaginal microbiome of healthy dairy cattle suggests an important role in maintaining reproductive success [15,16,17,19]. *Clostridium* (within phylum Firmicutes) has also been identified as a common genus present in the vagina, most likely due to its close proximity to the rectum [14,20].

Few studies utilizing culture-based techniques have indicated the presence of genera such as *Truperella* and *Escherichia*, especially in cows with reproductive infections. However, this technique is inadequate on its own due to the specific growing conditions required for various bacteria. Identification of commensal or pathogenic bacteria throughout various timepoints in the reproductive life of dairy cattle is imperative to understanding reproductive success and failure. Differences in vaginal microbiomes between primiparous and multiparous cattle have been reported [17]. This is most likely a result of previous calvings and the subsequent vulnerability of the reproductive tract allowing for the introduction of new bacteria to a compromised system. Certain bacteria have been associated with reproductive infections. Specifically, the genera *Histophilus* and *Mogibacterium* present prior to calving have been associated with cows that later developed endometritis during the postpartum period [19]. One study determined that cows with lower alpha diversity (Shannon index) at calving were associated with the development of metritis, indicating closely related types of bacteria that could influence the onset of reproductive infections [20]. Additionally, this study found that changes in phyla Fusobacteria and Bacteroidetes were indicative of postpartum reproductive diseases including metritis and clinical endometritis, as well as poor reproductive success [20]. An additional study has reported that the increased diversity levels in the vaginal microbiome prior to calving are related to a decreased incidence of reproductive diseases during the postpartum period, along with a greater abundance of the phylum Firmicutes. Oppositely, lower bacterial abundance and diversity are associated with the onset of reproductive diseases including endometritis. This is important to note, as the vaginal and uterine microbiomes have a short period of time post-calving where they coalesce. This roughly seven-day period leads to alterations in both the uterine and vaginal microbiomes that allow bacteria to travel between the two organs and influence the onset of disease [21]. Vaginal microbiome genera that have been associated with endometritis have been reported to include *Bacteroides*, *Helococcus*, *Prevotella*, *Porphyromonas*, and *Fusobacterium* [18,20,21].

#### 3.1.2. Uterine Microbiome

Initial research proposed the dogma that the uterus was a sterile environment in order to establish and maintain a pregnancy. However, recent research into microbiomes of the reproductive tract, including the uterus, is challenging this theory. Based on recent studies, the core phyla that have been identified in the uterine microbiome in both heifers and cows include Bacteroidetes, Proteobacteria, Firmicutes, and Actinobacteria, along with Fusobacteria and Tenericutes, similar to other portions of the cattle reproductive tract [14,22,23,24]. Across studies, common genera in the uterine microbiome include *Fusobacterium*, *Porphyromonas*, and *Bacillus* [21,25,26,27]. The identification of these bacteria in dairy cattle has come from many studies that focus on the uterine microbiome during periods of infection utilizing both culture-based and 16S rRNA sequencing technologies.

Evaluations of the uterine microbiome in cattle continue to focus on bacteria associated with infection that commonly occur during the postpartum period. One study noted that cows with purulent vaginal discharge, a sign typically indicative of clinical endometritis, had an increased abundance of Bacteroidetes and Fusobacteria [28]. The presence of Fusobacteria and Bacteroidetes has also been established in cows with metritis, a severe infection of the uterine layers. Specifically, Fusobacterium and Bacteroides, along with *Helococcus* and *Campylobactera*, could also be indicative of metritic cows [22,26,27]. Together, *Trueperella pyogenes* and *Escherichia coli* have also been known to work in promoting uterine infections, while separately having the potential to influence the severity of the disease [19,26,27]. One study evaluating metritis found that *T. pyogenes* (also known as *Corynebacterium pyogenes*) was present in all cow samples, and therefore may be more indicative of a pathogenic bacterium for endometritis. *E. coli*, also a common indicator of uterine infection, was also observed and could lead to metritis or other uterine diseases if in greater abundance immediately post-calving. Similar to culture-based studies, *Fusobacterium necrophorum* and *Prevotella melaninogenica* were also identified as potential indicators of metritis along with newer identifications including *Bacteroides pyogenes*, *Porphyromonas levii*, and *Helcococcus ovis* [26]. The combination of *Trueperella* and *Helococcus* was found in another study in uterine microbiome samples of cows experiencing endometritis [19].

Identifying the uterine microbiome under conditions other than infections is also imperative to continue understanding reproductive tract microbiomes and can be accomplished through 16S rRNA sequencing. The postpartum period is also known for negative energy balance (NEB) in dairy cattle, as they put more energy into producing milk than they are able to consume in food. One study evaluating the uterine microbiome in cows experiencing NEB versus those that were not found differences in phyla dominance from Bacteroidetes and Fusobacteria versus Proteobacteria and Firmicutes. These continued to change across timepoints, with shifts towards Actinobacteria, Proteobacteria, and Cyanobacteria later in NEB [23]. Another study evaluating differences between parities did note that the relative abundance of various genera depended on parity status. Primiparous cattle tended to have greater diversity when compared to multiparous cows [25]. Together, these findings emphasize the importance of understanding the differences between infected and healthy uterine microbial communities. However, further research is needed to continue determining the mechanisms behind these changes as well as the status of a healthy uterine microbiome throughout seasons, breeds, and changes in cow reproductive status.

#### 3.1.3. Placental Microbiome

As previously mentioned, the dogma was that the uterus must be a sterile environment for both the establishment and maintenance of pregnancy across species. Currently, there are opposing theories regarding the existence of a placental microbiome in humans. While certain studies suggest that any bacteria identified are from contamination during post-labor collections and/or sequencing, others indicate that placental microbiomes do exist pre-labor and appear similar to the oral microbiome of the fetus [29,30,31,32]. While placental microbiome identification in dairy cattle has not been extensively studied compared to that in humans, similar research utilizing 16S rRNA sequencing in placental microbial transfer to the gastrointestinal tract (GIT) of neonates has recently become a large focus.

With research methods shifting from culture-based to sequencing techniques, one study suggests that the phyla present in amniotic fluid are different from those in both ruminal and cecal fluids as well as meconium in dairy calves. The most prevalent phylum in the amniotic fluid was Bacteroidetes, while the abundance of Bacteroidetes in the calf GIT samples was low [32]. Another study focusing on locational differences between placental microbial communities noted a low abundance of bacteria that was centered at the placentomes towards the chorioallantoic membrane. At this site, which is close to the maternal/fetal interface, the presence of bacteria could further indicate the potential passive transfer of microbes from the maternal blood supply to the fetus [33]. On the other hand, a different study noted that bacterial richness and evenness (alpha diversity; Chao1 and Shannon index) of the amniotic fluid, intercotyledonary placenta, and placentomes were all similar [24]. Across studies, it appears that the dominant phyla present in placental tissues are Bacteroidetes, Firmicutes, Proteobacteria, and Actinobacteria [24,32,34]. The similarity of identifying Bacteroidetes between these studies suggests that it makes up a distinguishing portion of the placental microbiome, and that this microbiome does exist in dairy cattle. Specifically, low levels of unique operational taxonomic units between samples indicate that the placental microbiome is very conserved across each distinct portion [34]. These findings suggest a divergence from the initial dogma of a sterile uterus and indicate the need for further investigation into the identification of the dairy cattle placental microbiome and its impacts on pregnancy and the establishment of the offspring microbiomes.

#### 3.1.4. Colostrum and Milk Microbiome

Insights into the milk microbiome in dairy cattle have been increasingly studied over the years as mastitis continues to be a prevalent issue across herds. In studies utilizing both culture-based approaches and 16S rRNA sequencing, it has been established that the core phyla present within milk and colostrum samples include Firmicutes, Proteobacteria, Bacteroidetes, and Actinobacteria [35,36,37,38,39]. More specifically, common genera that have been identified in milk microbiome samples include *Corynebacterium*, *Escherichia*, *Pseudomonas*, *Lactococcus*, *Methylobacterium*, *Streptococcus*, *Acinetobacter*, and *Staphylococcus* [36,40,41,42]. In the evaluation of differences between healthy milk and infected milk samples, studies indicate that mastitic milk trends towards uneven distributions of taxa. *Streptococcus*, *Staphylococcus*, and *Escherichia* appear to consistently be high in abundance in infected milk samples, with other genera, including *Pseudomonas*, *Corynebacterium*, *Bacillus*, and *Actinobacter*, varying between richness in healthy or infected samples [36,37,40,42]. Furthermore, the severity of mastitis (i.e., types of bacteria causing mastitis) has been shown to impact reproductive performance, pregnancy per first AI, in dairy cows. Mastitis caused by major pathogens including *Staphylococcus aureus*, *Streptococcus agalactiae*, *Escherichia coli*, *Klebsiella*, *Mycoplasma*, *Streptococcus uberis*, or *Streptococcus dysgalactiae* is associated with lower pregnancy per first AI and greater incidence of pregnancy loss [43]. Specifically, mastitis caused by Gram-negative bacteria including *E. coli* or *Klebsiella* was associated with the lowest pregnancy per first AI and the greatest incidence of pregnancy loss when compared to either cows without mastitis or cows diagnosed with mastitis caused by Gram-positive bacteria [43].

Microbial communities present in colostrum have also been a growing focus in microbiome research. Colostrum, the first milk post-calving, plays a crucial role in the passive transfer of immunity and nutrients from the dam to the calf [44]. Currently, the majority of studies are aimed at evaluating relationships between the colostrum microbiome and the gut microbiome of calves to determine the extent to which colostrum influences bacterial prevalence within the digestive tract. From one study, it appears colostrum plays a role in developing the early gut microbiome before the gut begins to facilitate its own bacterial communities, as indicated by similarities between colostrum and first fecal samples. The genera that were commonly identified in their colostrum samples were *Corynebacterium*, *Acinteobacter*, *Enterobacter*, and *Streptococcus* [45]. Another study that evaluated differences in bacteria abundance within colostrum between multiparous and primiparous cows identified *Staphylococcus*, *Prevotella*, *Pseudomonas*, *Actineobacter*, *Fusobacterium*, and *Bacteroides* as genera present within all of the samples. Multiparous cows specifically had a higher abundance of *Staphylococcus*, *Fusobacterium*, *Acinetobacter*, and *Bacteroides* compared to their primiparous counterparts. Similar to the milk microbiome, Firmicutes, Proteobacteria, Bacteroidetes, and Actinobacteria appear as the common phyla across colostrum samples, indicating that while the richness of core genera changes throughout milk production, the overall phyla present continue to persist [44]. With studies reporting various changes in genera abundance dependent on the type of milk samples, more research utilizing 16S rRNA identification will be beneficial to continue elucidating the milk and colostrum microbiome during periods of infections, between breeds, across seasons, and between parities.

### 3.2. Male Reproductive Microbiome

#### Semen Microbiome

There are minimal studies focusing on dairy bull seminal microbiome, and the majority of the literature evaluates bacterial colony cultures. From these studies, it has been established that bacterial loads vary significantly between breeds, where Jersey bulls have greater bacterial loads, followed by crossbreds and the zebu cattle breeds including Gir and Red Sindhi [46]. Research has also evaluated differences in bacterial microbiomes between fresh and frozen semen. Semen is frozen with an extender that contains antibiotics aimed at limiting bacterial contamination. However, studies evaluating bacterial presence during thawing found that these antibiotics do not fully eliminate bacteria, and bacteria can be detected in all semen straws by 96 h post-thawing [46,47,48]. The presence of bacteria within the seminal microbiome can decrease dairy bull fertility by causing changes in sperm motility and morphology and lowering sperm DNA integrity. These bacteria also have the potential to transmit diseases to females in natural mating and inhibit fertilization, thereby lowering pregnancy rates [47,48,49]. Yet, the identification of specific bacteria that could be contributing to the inhibition of male and/or female fertility requires further investigation.

Recent research has shifted towards the use of PCR targeting 16S rRNA for bacterial identification that could elucidate specific bacteria present in the dairy bull seminal microbiome that aid or inhibit fertility. Common phyla present in bull seminal microbiomes across studies include Firmicutes, Proteobacteria, Fusobacteria, Actinobacteria, and Bacteroidetes [47,50]. Genera within Firmicutes and Fusobacteria have been associated with the preputial microbiome and could indicate poor environmental conditions or a lack of hygiene during collections [50,51,52]. Additional genera *Lawsonella*, *W5053*, *Campylobacter*, *Hemophilus*, and *Mycoplasma* have also been found in seminal microbiomes in dairy bulls [51]. Together, this indicates that the antibiotics present in frozen semen may not be sufficient to prevent the growth of bacterial communities, and additional research into identifying the dairy seminal microbiome is needed.

## 4. Beef Cattle

### 4.1. Female Reproductive Microbiome

#### 4.1.1. Vaginal Microbiome

Reproductive inefficiency can have a significant impact on the success of a cow–calf beef operation. Given the relationship between vaginal microbiota and fertility in women, many studies have sought to determine the vaginal (and other reproductive tissues) microbiome in beef cattle. Similar to dairy cattle, dominant phyla found within the vagina of beef cattle are Bacteroides, Firmicutes, Proteobacteria, and Actinobacteria [14,53,54,55]. Pickett et al. [53] also found the phyla of Tenericutes and Spirochaetes within flush samples. However, individual studies varied by specific relative abundances of various phyla. Bacterial communities in the vagina that are found to be more diverse have been associated with a normal or disease-free reproductive tract in beef cattle [14]. A more diverse vaginal environment can potentially be attributed to the proximity to the external environment [14,56]. The genera *Pedobcter, Paraprevotella*, and *Porphyromonas* have been associated with both the bovine rumen and the vaginal environment. *Pedobacter* and *Paraprevotella* are bacteria found within fecal excretions. Shifts in the bacterial abundances of the reproductive tract could be attributed to levels of feed concentrate in the rumen and fecal matter when excreted [53].

In cattle being synchronized for artificial insemination, there was a decrease in Firmicutes in the days leading up to artificial insemination [57]. Under the phylum Firmicutes, the genus *Peptoniphilus* was found to be more abundant in cattle that were not able to establish a pregnancy [58]. Ault et al. [57] and Messman et al. [55] speculate that the fluctuating bacterial communities could be associated with the hormonal shifts of the estrous cycle. Shifts of progesterone concentrations were correlated to the relative abundance of the phyla Firmicutes and Proteobacteria. Specifically, as progesterone concentrations declined, there was an increase in Firmicutes, whereas when progesterone levels increased, there was an increase in Proteobacteria [57]. In comparison to the uterine environment, the vaginal microbiome has been characterized as having more diversity and a greater bacterial species number [56]. In humans, vaginal pH is more acidic (pH of 5.0–5.5) due to the presence of *Lactobacillus*, whereas a more neutral pH environment (pH of 7.0–7.5) in the vagina of beef cattle can be, in part, attributed to lower abundances of *Lactobacillus* [54,55]. Shifts in pH can impact the relative abundances that can colonize the reproductive environment, and shifts in pH could be due to changes in reproductive hormones such as progesterone and estradiol. As previously stated, progesterone appears to be associated with shifts in the relative abundance of bacteria, whereas there appears to be a minimal influence of estradiol on the vaginal microbiome in beef heifers [55]. There is a gap in the literature regarding how estrus synchronization tools (for example, progesterone-releasing devices) can affect the vaginal microbiome. However, some studies have shown that improper handling of such tools could be a cause of the outside introduction of bacteria into the vagina, although this does not appear to negatively impact fertility [59].

#### 4.1.2. Uterine Microbiome

Dominant phyla found within the uterus of beef cattle are Bacteroides, Firmicutes, Proteobacteria, Actinobacteria, and Tenericutes [14,57,60]. Specific bacterial communities can potentially alter fertility outcomes within the uterus, but this has been minimally investigated in beef cattle. The genera of bacteria found in females that were not able to establish a pregnancy were *Blautia, Butyrvibrio*, and *Natronincola,* which are under the phylum Firmicutes. High relative abundances of Firmicutes have been characterized in cattle that develop endometritis, supporting the hypothesis of its adverse effects [58]. As previously mentioned in the vagina, shifts in hormonal concentrations may cause the uterine bacterial communities to shift either towards a pathogenic or commensal environment and can impact reproductive success [56,60]. Within the uterus, greater endogenous progesterone concentrations result in greater bacterial diversity [60]. Prior to breeding, the genus *Corynebacterium* was found in cows that were not able to become pregnant [57]. As mentioned in the dairy section, this genus of bacteria has been associated with uterine disease [20]. Compared to the vagina, a less diverse uterine environment could possibly be attributed to restricted access from an external environment [14,57].

#### 4.1.3. Placental Microbiome

The placenta in mammalian animals is essential because it contains the membranes and blood vessels that connect the fetus to the blood supply of the mother. Still, the placental microbiome in beef cattle has not been studied extensively and is poorly misunderstood. Hummel et al. [61] posited that restricting feed intake will cause a less diverse rumen microbiome which may alter the placental microbial composition. The rumen of those feed-restricted cows was less rich and diverse than those females that were not restricted. These changes within the rumen could influence the fetal bacterial communities. The gastrointestinal community shares similarities between it and the cotyledon, the site of gaseous exchange, where a method of bacterial advancement may occur [61]. In newborn calves, it has been noted that amniotic fluid and meconium are similar, which can be surmised by the fact that amniotic fluid is ingested during development which in turn introduces the fluid to the fetal gut [62].

### 4.2. Male Reproductive Microbiome

#### Penile and Semen Microbiome

Artificial insemination was introduced to the livestock industry to reduce the spread of disease and infection across the herd. In the human literature, Mandar et al. [63] indicated that infertility may not only be associated with the female reproductive tract but could also be attributed to the seminal fluid that is being introduced during copulation. Pathogenic bacteria, such as *Histophilus*, have been found on the prepuce of bulls and have been implicated in disease [64]. Similar to the vagina of the cow, the phyla of Firmicutes, Fusobacteria, Bacteroidetes, Proteobacteria, and Actinobacteria were found within the prepuce and penis of the bull and within the seminal fluid [64,65,66]. The most abundant genera identified were *Escherichia*, *Bacteroides*, *Corynebacterium*, and *Streptococcus* on the surface of the penis [64]. Within seminal samples, the most common genera were *Bacteroides*, *Escherichia*, and *Gemella*. Bulls deemed unsatisfactory by a breeding soundness evaluation had lower abundances of *Escherichia* and *Staphylococcus* when compared to satisfactory males [65]. Luecke et al. [67] state that there may be a complementary environment between the male and female microbiome as semen travels through the tract. Transmissions of microbes could be a potential factor that influences homeostasis within the female reproductive tract and allow for the successful establishment of pregnancy.

## 5. Sheep and Goats

### 5.1. Female Reproductive Microbiome

#### 5.1.1. Vaginal and Uterine Microbiome

Earlier work utilizing culture-based methods identified commonalities in vaginal microflora present in ewes and does. Predominant bacterial genera and species present include *Staphylococcus*, *Corynebacterium*, *Bacillus*, *Streptococcus,* and *Escherichia coli* [68,69,70,71]. One study specifically evaluated bacterial populations in association with embryonic death and found that *Pasteurella multocida* was isolated in all ewes in which embryo death was observed [69]. This bacterium is pathogenic for many mammals and birds; specifically for sheep and goats, it is one of the main contributors to respiratory diseases such as pneumonia [72,73].

With advancements in 16S rRNA sequencing, some work has further investigated the vaginal microbiome in small ruminants, although not to the extent of that in beef and dairy cattle. Using this technology, it was revealed that the vaginal microbiome differs from culture-based results. Characterizing the vaginal microbiome, ewe vaginal samples exhibited less alpha diversity compared to cow vaginal samples, though both were more diverse when compared to the human vaginal microbiome. In the ewe, the most abundant genera were *Aggregatibacter* and *Streptobacillus*, with bacteria previously identified using culture-based methods being present in low abundance [54]. Interestingly, Serrano and colleagues [74] also found *Streptobacillus* in ewe vaginal samples, with the use of the antibiotic framycetin significantly reducing its abundance. Follow-up studies sought to identify differences in vaginal microbiota between pregnant and nonpregnant ewes. Bacterial genera that were more abundant in nonpregnant ewes include *Histophilus*, *Actinobacillus*, *Streptococcus*, *Mannheimia*, and *Fusobacterium* [74,75]. Specifically, *Histophilus* was found in both studies and belongs to the Pasteurellaceae family, which is a known livestock pathogen causing respiratory disease and reproductive disorders such as abortion and epididymitis [74,75,76]. To date, only one scientific abstract has characterized uterine bacteria in sheep, with the genus *Finegoldia* being the greatest in nonpregnant ewes [77]. Interestingly, a species within this genus, *Finegoldia magna*, is an opportunistic pathogen and has been associated with bacterial vaginosis in humans [78]. Additionally, no studies have characterized the goat vaginal microbiome using 16S rRNA sequencing.

#### 5.1.2. Placental Microbiome

Culture-based methods have been used to identify certain pathogenic bacteria that are associated with placentitis (i.e., inflammation of the placenta) and abortion in sheep and goats. These largely include *Brucella*, *Campylobacter*, *Chlamydia*, *Listeria*, and *Coxiella* [79,80,81]. To date, no studies have directly characterized either the sheep or goat placental microbiome using 16S rRNA sequencing; however, a couple of studies have identified the existence of a microbiome in the prenatal lamb and kid gut suggesting microbial seeding occurring in utero [82,83].

#### 5.1.3. Colostrum and Milk Microbiome

Similar to dairy cattle, mastitis is inflammation of the mammary gland caused by pathogenic bacteria and results in production losses for both dairy and meat sheep and goats. However, it is important to note that there are differences in the pathogenic bacteria that cause mastitis in dairy cattle versus small ruminants [84]. In sheep without mastitis, prevalent microbiota in milk includes *Staphylococcus* (12.3% to 14.1%), *Lactobacillus* (12.2%), and *Corynebacterium* (4.3% to 9.7%) [85,86]. Sheep diagnosed with subclinical mastitis also had this core microbiota within milk samples, yet bacteria genera also identified included *Escherichia* (0.8% to 2.1%), *Mannheimia* (1.9%), and *Enterococcus* (2.4%) [85,86,87]. These findings are similar to those of cultured-based studies in which several bacteria were found to cause mastitis or subclinical mastitis in sheep, including *Escherichia coli*, *Mannheimia*, *Enterococcus*, and coagulase-positive staphylococci [87]. In goats, the phylum Proteobacteria is dominant in colostrum, and this profile shifts throughout lactation, with the phylum Actinobacteria being dominant in milk [88]. Utilizing 16S rRNA sequencing, *Enterococcus* spp., *Staphylococcus* spp., and *Bacillus* spp. were found to be prevalent in milk samples from goats without mastitis [87]. Specific bacteria responsible for mastitis in goats include *Mycoplasma*, *Escherichia*, and *Enterococcus*, and this is similar for both 16S rRNA sequencing and culture-based methods [89]. For both sheep and goats, as the severity of mastitis increases (subclinical to gangrenous mastitis), alpha diversity metrics such as Chao1 and Shannon decrease, which potentially indicates a specific group of bacteria responsible for the progression of the disease [85,89].

### 5.2. Male Reproductive Microbiome

#### Semen Microbiome

Using culture-based techniques, it was found that ram semen consists of similar bacteria to those of the ewe vagina. Some of the most prevalent bacteria include *Escherichia coli*, *Staphylococcus*, *Streptococcus*, and *Bacillus* [90,91,92]. Interestingly, these bacteria as well as *Histophilus somni* were isolated from semen samples collected from rams with normal (≥70% motility) and reduced (<70% motility) sperm quality, indicating that these bacteria are not necessarily detrimental to sperm quality [90]. *Mannheima haemolytica* was only isolated from semen samples from rams with reduced sperm quality, and this bacterium is a known pathogen in rams [90,93]. Only two studies have used 16S rRNA sequencing on semen from rams and bucks. In rams, the most abundant bacterial genera were *Corynebacterium* and *Pseudomonas* [74]. In bucks, ejaculates were collected during the breeding (November and December) and non-breeding (March) seasons; microbial community structure (beta diversity metrics) differed, with *Sphingomonas* and *Halomonas* being more abundant during the breeding season, and *Sphingomonas* was positively correlated with sperm quality whereas *Faecalibacterium* was associated with poor sperm quality [94].

## 6. Swine

### 6.1. Female Reproductive Microbiome

#### 6.1.1. Vaginal Microbiome

Early research utilizing culture-based techniques isolated bacterial colonies from vaginal samples in normal healthy sows [95]. These included *Streptococcus*, *Escherichia coli*, *Staphylococcus*, *Corynebacterium*, and *Micrococcus*. More recently, 16S rRNA sequencing has allowed for further classification of the bacterial genome. Many swine microbiome studies have focused on the GIT but have suggested there are relationships between microbiota present in the gut and reproductive tract [96]. Zhang et al. [97] identified that Firmicutes, Bacteroidetes, and Spirochaetes were the most abundant phyla in the GIT of sows, while Firmicutes, Proteobacteria, and Bacteroidetes were the most abundant in the vagina. Vaginal microbial diversity is important for understanding the role of the microbiota in reproductive tract health and developing new ways to prevent and treat infections by modulating the microbial community. Comparing bacterial abundance between healthy and unhealthy (i.e., diagnosed with endometritis) sows, the phylum Firmicutes was more abundant in the vagina of healthy sows, while Proteobacteria and Bacteroidetes were more abundant in endometriotic sows. At the genus level, *Bacillus* and *Paenibacillus* were more abundant in healthy sows, while *Escherichia-shigella* and *Bacteroides* were more abundant in sows with endometritis [98]. This is consistent with another study in which sows at high risk of pelvic organ prolapse had a reduced abundance of Firmicutes and a higher abundance of Bacteroidetes and Proteobacteria [99].

#### 6.1.2. Uterine Microbiome

The role of the vaginal microbiota has been studied to a greater extent when compared to the uterine microbiota. Recently, 16S rRNA gene sequencing identified several phyla present in the uterus that differ between preimplantation (day 11 of gestation) and implantation (day 15 of gestation) periods of gestation in healthy gilts. Firmicutes were the most abundant across both periods, while Bacteroidetes were more abundant during the implantation period compared to the preimplantation period [100]. Further research examined bacterial communities during later stages of gestation and found stark differences compared to earlier stages of gestation. At day 60 of gestation, Firmicutes and Fusobacteria showed a lower relative abundance in the uterus compared to all other reproductive tissues, while Proteobacteria was the most abundant phyla in the uterus compared to other tissues. At the genus level, *Lactobacillus*, *Porphyromonas*, and *Oscillospira* were found to be less abundant in the uterus compared to the vagina at day 60 of gestation, while *Acinetobacter*, *Streptococcus*, and *Clostridium* were more abundant in the uterus compared to the vagina [100].

#### 6.1.3. Placental Microbiome

The microbiome of the pig placenta is mostly unknown, with minimal insight regarding bacteria colonization. Through early culture-based methods, it was observed that there was a higher relative abundance of *Rhodococcus* and *Actinomadura madurae* in the chorioallantois of aborted fetuses [101]. These two genera are Gram-negative bacteria commonly associated with placentitis in porcine. Additionally, recent work has shown Firmicutes to be the most abundant phylum in allantoic fluid of clinically healthy gilts. The genus *Lactobacillus* was also observed to have a higher relative abundance in allantoic fluid compared to the rest of the reproductive tract [100]. Further research is required to fully understand the role microbiota plays in placental development and maintenance throughout gestation in swine.

#### 6.1.4. Colostrum and Milk Microbiome

The origin of the sow colostrum microbiome is unknown; however, two theories have been developed. Previous studies have found evidence to support that the sow’s teat and surrounding skin may be the origin of the colostrum microbiome as many similar bacteria species have been found on the teat and within the colostrum microbiota [102]. A study conducted by Kemper and Preissler [102] observed species from the families Staphylococcaceae, Streptococcaceae, and Enterobacteriaceae in both bacterial communities of the mammary gland tissue and colostrum microbiome. In contrast, current research is working to establish the entero-mammary pathway for bacteria colonization, which may confirm the theory that the sow’s gastrointestinal microbiome may influence her colostrum [103].

Colostrum is an essential component in a neonate’s life as it is the optimal source of nutrition for development [104,105]. However, colostrum impacts the offspring in other important aspects such as the development of the piglet’s innate immune system and gastrointestinal microbiome [103,106]. According to Liu et al. [106], colostrum largely contributes to the development of a piglet’s gastrointestinal microbiome, with approximately 90% of the bacteria within the small and large intestine originating from colostrum during the piglet’s first 35 days of life. This idea was supported by Maradiaga et al. [107], who found that the bacterial community of a piglet’s gastrointestinal tract was highly correlated with the sow’s colostrum microbiota.

The phyla Firmicutes and Proteobacteria dominate the sow milk microbiome, with Bacteriodetes, Actinobacteria, Fusobacteria, and Tenericutes being observed at lower abundances. *Lactobacillus reuteri* was found to have the greatest relative abundance compared to other bacterial species within the sow milk microbiome [104]. Genera including *Bacteroides*, *Ruminococcus*, *Blautia*, and *Bifidobacterium* are all bacteria associated with the pig GIT and have also been present in the sow’s milk microbiome [102,104].

Little research has focused on the bacterial diversity of milk; however, recent research has indicated that it may change over time. For example, alpha diversity was greater on day 0 (i.e., colostrum) than on days 1 and 3. When looking at beta diversity, the milk microbiota changes during lactation, with transitional and milk samples being more similar than colostrum samples [104].

### 6.2. Male Reproductive Microbiome

#### Semen Microbiome

The relationship between the boar semen microbiome and reproductive performance remains uncertain, despite the frequent presence of bacterial contaminates being well documented using culture-based methods. Interestingly, the seminal bacterial composition differs between neat and extended semen. Common bacterial communities in neat, non-processed semen include *Escherichia coli*, *Pseudomonas*, *Staphylococcus*, *Klebsiella*, *Proteus*, and *Citrobacter* [108,109]. In contrast, several studies indicated that the Enterobacteriaceae family had a high relative abundance in extended boar semen samples. Frequently isolated bacterial species within the Enterobacteriaceae family include *Serratia marcescens*, *Morganella morganii*, *Proteus mirabilis*, *Escherichia coli*, *Klebsiella*, and *Enterococcus* [108,109,110].

Using 16S rRNA sequencing, several studies have observed Proteobacteria, Firmicutes, Actinobacteria, and Bacteroidetes as the most prominent phyla within the boar seminal microbiome [111,112,113]. These same phyla were also found using culture-based techniques. Interestingly, many of these bacteria can either be commensal or pathogenic due to their effects on sperm cells. In the phylum Firmicutes, the genus *Clostridium* was more abundant in semen with greater sperm cell agglutination and reduced sperm motility [113]. This genus has been considered pathogenic due to its association with reduced sperm quality based on previous research using culture-based practices [114,115,116]. Commensal bacteria have also been observed in the boar semen microbiome using 16S rRNA sequencing. Several genera including *Streptococcus*, *Mannheimia*, *Psychrobacter*, *Moraxella*, and *Brennera* were present in semen with acceptable sperm quality; therefore, they may not be pathogenic [113,117]. Using sequencing technology, it was found that bacterial orders Clostridiales, Lactobacillales, and Bacteriodales are more abundant in semen when boars are housed on slatted floors than on sawdust floors. In comparison, when boars are housed on sawdust floors, Enterobacteriales and Rhizobiales are more abundant in semen [117].

Alpha diversity metrics such as Chao1 and Shannon’s are used to measure the evenness and richness of a bacteria population within a sample. Some studies’ findings suggest that a reduced alpha diversity bacterial community is more favorable for fertility [112,113]. Further, greater alpha diversity was exhibited in ejaculates collected in the summer months, which had a lower fertilizing capacity than winter ejaculates [112]. Alpha diversity can also differ by geographical location as boars from France have greater richness and evenness within their semen than boars from the United States [117]. Overall, further research investigating the boar semen microbiome using 16S rRNA sequencing must be conducted to understand the dynamics between sperm cells and the bacterial population that resides within boar semen.

## 7. Equines

### 7.1. Female Reproductive Microbiome

#### 7.1.1. Vaginal and Uterine Microbiome

Using 16S rRNA sequencing and culture-dependent methods in Arabian mares, it was found that the stage of the estrous cycle appears to be associated with shifts in bacterial communities within the vagina [118]. The dominant phyla in estrus (E) and diestrus (D) were Firmicutes (E: 32.03%, D: 31.51%) and Bacteroidetes (E: 31.98%, D: 31.15%) within the mares. The relatively similar microbiomes detected between the two phases of the estrous cycle indicate that healthy-appearing mares have a relatively stable profile of bacterial communities. Two major genus-level components consisted of *Porphyromonas* (E: 13.90%, D: 16.94%) and *Campylobacter* (E: 7.46%, D: 10.93%) [118].

In recent years, the proposed idea that the uterus remains a sterile environment for the developing fetus has been challenged. To determine the identity of unculturable bacteria, 16S rRNA bacterial sequencing identification has been implemented to allow for the characterization of the uterine microbiome and other reproductive tissues. When evaluating the uterine microbiome of mares located in either the United States or Australia, bacterial genera vastly differed, which could indicate a geographical influence on the microbiome [119]. The uterus remains susceptible to infection by opportunistic bacteria such as *Streptococcus zooepidemicus*, *Staphylococcus aureus*, and *Escherichia* (commonly identified using culture-based methods) resulting in endometritis, which can lead to severe economic loss due to associated infertility [120].

Understanding the bacterial differences in healthy and endometriotic equids (i.e., donkeys) would further aid in the understanding of endometritis. The similarity of the composition of bacterial communities between the endometrial and vaginal swabs of donkeys indicates an influence of the vaginal microbiome on the uterine microbiome. Anaerobic bacterial changes in the phyla and family levels were noted in jennies that were diagnosed with endometritis in both their endometrial and vaginal microbiomes. Specifically, bacterial families Ruminococcaceae and Lachnospiraceae were prevalent in both vaginal and endometrial swabs of jennies diagnosed with endometritis [121]. Developing an understanding of the healthy components of a vaginal microbiome in addition to the effects of the bacteria within the vagina on the further reproductive tissues such as the uterus is key to improving fertility in equids.

#### 7.1.2. Placental and Lactational Microbiome

The identification of bacteria specifically within portions of the placenta, including the allantois and amnion, has been a small focus in the larger goal of aiding microbiome identification within the GIT of foals. Using 16S rRNA sequencing, one study evaluated both the gravid and non-gravid portions of chorioallantois tissues in healthy mares and found that Firmicutes, Proteobacteria, and Bacteroidetes were the most abundant phyla, as well as high levels of the genera *Clostridium* and *Moraxella* [122]. Other research evaluating amniotic fluid in comparison to sections of foals’ GIT found that Bacteroidetes, Actinobacteria, and Firmicutes were the most prevalent phyla within the amniotic fluid, similar to that of the chorioallantois [123]. At the genera level, another study evaluating the establishment of the foal’s hindgut microbiome reported *Sphingomonas*, *Pseudomonas*, and *Staphylococcus* as some of the bacteria present within the horse amniotic fluid [124]. With the use of 16S rRNA sequencing technology and previous research in mind, it is imperative to continue identifying the mare placental microbiome in various reproductive states to further understand its implications on reproductive success.

With the purpose of understanding the establishment of the foal’s hindgut microbiome, 16s rRNA sequencing was used in the comparison between the foal meconium and feces against the mare’s feces, colostrum, and amniotic fluid as indicated previously. Samples of the foals’ feces for comparison to the milk and colostrum microbiome were taken on days 1, 3, 5, 7, and 10 for analysis. When analyzing the influence of the mare’s colostrum and milk on the establishment of the foal’s gut microbiome, day 3 indicated an authority of the bacterial meta-community of the milk and feces. There was an indicated loss of the meconium community in conjunction with the milk intake. However, with continued collection, the milk microbiome’s resemblance to the microbiome of the foals’ feces began to decrease as the mare and foal microbiome communities began to converge [124].

### 7.2. Male Reproductive Microbiome

#### Semen Microbiome

Evaluating the influence of bacteria on the survivability of spermatozoa in cooled-stored stallion semen will allow for the characterization of a healthy microbiome in stallions as it pertains to fertility. The composition of the semen microbiome allows for a better understanding of the specific antibiotic that should be added to the extender to eliminate pathogenic bacteria. Using culture-based methods, it was identified that *Streptococcus equisimilis* and *Pseudomonas aeruginosa* reduce sperm motility and velocity, and these effects were not improved by the addition of the antibiotic gentamicin [125]. Using 16S rRNA sequencing, 69 bacterial families were detected between 12 healthy, fertile stallions, with Porphyromonadaceae (33.18%), Peptoniphilaceae (14.09%), Corynebacteriaceae (11.32%), and Prevotellaceae (9.05%) being the most prominent families [126]. An additional study with seven stallions found a wide distribution of bacterial genera within semen samples, with *Porphyromonas*, *Corynebacterium*, and *Finegoldia* being the most prevalent [127]. However, neither of these studies that used 16S rRNA sequencing related certain bacterial communities with sperm fertility parameters such as sperm motility.

## 8. Conclusions

As shown in Table 1, there is a wide range of pathogenic bacteria that impact reproduction in domestic livestock. Many of these have been identified using culture-based methods; however, advancements in 16S rRNA gene amplicon community sequencing have allowed for a more precise classification of bacteria as either pathogenic or commensal within the reproductive tract. In humans, utilizing 16S rRNA sequencing for the HMP identified that *Lactobacillus* accounts for over 90% of the vaginal microbiome in healthy women. Moreover, using 16S rRNA sequencing in reproductive tissues and samples from livestock animals has helped to identify factors such as reproductive hormones and disease that could be altering or associated with shifts in bacterial populations. While there are limitations with 16S rRNA sequencing, further investigation is needed to conclusively identify commensal bacteria within the reproductive tract of domestic livestock. Continued understanding of reproductive microbiota will aid in the development of management and therapeutic strategies to improve reproductive efficiency in domestic livestock animals.

## Figures and Tables

**Table 1 animals-13-00485-t001:** Summary of pathogenic bacteria in reproduction for domestic livestock.

	Culture-Based Method	16S rRNA Sequencing	References
Dairy Cattle			
Vagina	*Trueperella* and *Escherichia* associated with metritis	Fusobacteria and Bacteroidetes phyla with metritis and endometritis	[19,20]
Uterus	*Trueperella pyogenes* and *Escherichia coli* associated with metritis	*Fusobacterium*, *Bacteroides*, *Prevotella*, and *Helococcus* with metritis and endometritis	[19,22,26]
Milk/Colostrum	*Escherichia* with mastitis	*Streptococcus, Staphylococcus,* and *Escherichia* with mastitis	[36,37,42]
Semen	*Myocplasma* infection of embryos and endometritis through natural mating	*Myocplasma* infection of embryos and endometritis through natural mating	[49]
Beef Cattle			
Vagina	*Mycoplasma* introduced via AI gun can cause abortion in cattle	Firmicutes and Proteobacteria shift with progesterone *Fusobacterium* found to be causative for morbidity and mortality *Bacteroidetes* associated with metritis	[57,65]
Uterus	-	*Corynebacterium* has been associated with infertility and uterine disease	[57,60]
Semen	*Histophilus* is found in mucosal membranes *Campylobacter fetus* causes campylobacteriosis	*Enterococcus* associated with poor semen quality	[64,65]
Sheep and Goats			
Vagina	*Pasteurella multocida* associated with embryonic death in ewes	Pasteurellaceae family (*Histophilus*, *Mannheimia*) abundant in nonpregnant ewes	[69,74,75,76]
Uterus	-	*Finegoldia* (Class Clostridia) abundant in nonpregnant ewes	[78]
Placenta	Placentitis and abortion caused by *Brucella*, *Campylobacter*, *Chlamydia*, *Listeria*, and *Coxiella*	-	[79,80,81]
Milk/Colostrum	*Escherichia coli*, *Mannheimia*, and *Enterococcus* cause mastitis or subclinical mastitis in sheep	*Mycoplasma, Escherichia,* and *Enterococcus* are bacteria responsible for mastitis in goats	[87,89]
Semen	*Mannheima haemolytica* isolated from semen with poor sperm quality from rams	*Faecalibacterium* (Class Clostridia) associated with poor sperm quality in bucks	[90,93,94]
Swine			
Vagina	-	Proteobacteria, Bacteroidetes, *Escherichia-shigella*, and *Bacteroides* with endometritis	[96,97,98,99]
Uterus	-	-	-
Placenta	*Rhodococcus* and *Actinomadura madurae* in the chorioallantois of aborted fetuses	-	[100,101]
Semen	Enterobacteriaceae family, Costridium perfringens, and Pseudomonas aeruginosa more abundant in semen with poor sperm quality and decreased longevity	Clostridium spp. and Streptococcus spp. associated with decreased sperm quality and leading to infections	[108,109,110,113]
Equines			
Vagina and Uterus	*Streptococcus zooepidemicus, Staphylococcus aureus,* and *Escherichia* isolated from mares with endometriosis	Ruminococcaceae and Lachnospiraceae families prevalent in donkeys with endometritis	[120,121]
Semen	*Streptococcus equisimilis* and *Pseudomonas aeruginosa* reduce sperm motility and velocity	-	[125]

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
