# Peer review of "Reproductive Microbiomes in Domestic Livestock: Insights Utilizing 16S rRNA Gene Amplicon Community Sequencing"

_animals, 2023, doi:10.3390/ani13030485_

Round 1

Reviewer 1 Report

I believe this is a great review, and information addressed in this review are lacking in the animal science field. However, since your goal is to address the new information that 16S rRNA sequencing provides in relationship to previous methods of bacteria culture, this review claims a section explaining how and why this technique allows for better understanding of the microbiome, as well as the disadvantages of this technique in relation to other metagenomic technologies.

Overall suggestions: 

The issue with culture systems is that you neglect the majority of bacteria that fail to grow under the adopted media conditions. While 16S rRNA sequencing allows for you to detect all members of the community, it also makes it hard to understand the role and contribution of every single member identified. Because of this, studies of the microbiome often use ecological parameters to study the microbiome, such as relative abundance of different phyla, bacterial richness, as well as alpha and beta diversity. These parameters of the microbiome are often addressed in this review, but explaining these ecological tools (perhaps under the same section you talk about the 16S rRNA sequencing) would also be beneficial to the common reader.

In addition, be cautious when using "healthy microbiome" or "healthy reproductive tract" as healthy can be subjective. It can either relate to a disease-free state, or to sound fertility and reproduction. While one might influence the other, the studies of the microbiome correlate the microbiome structure to either disease incidence and severity, or to reproductive success, and this should be clarified in this review.

Finally, I believe you can expand your conclusion with overall findings and what should be the next steps in the animal reproductive microbiome field. For as much as some species are more studied than others, and that results are somewhat confounding, there seems to be a clear difference in uterine and vaginal microbiome in all species, where the vaginal microbiome is more rich and diverse than the uterine microbiome (which is likely driven by the close proximity of the vagina to the environment and to the constant expose to feces). Furthermore, in all animals species addressed in the present review, the microbiome of the placenta was succesfully isolated, which disagrees with the believe that the placenta is a sterille environment, which could also be addressed in your conclusion.

More specific suggestions: 

Dairy Cattle Vaginal Microbiome

This section is a little confusing. You provide a lot of information, but there is no discussion of a central idea. 

My suggestion is that you take advantage of the very rich field of the human vaginal microbiome and make a comparison between what is well known in the human vagina VS. what might be happening in the bovine vagina given the data we have available.

In humans, bacteria from the genus lactobacillus are important to produce lactic acid, reducing the vaginal pH to an acidic environment, in which Lactobacillus bacteria are able to thrive while most other pathogenic bacteria cannot. Thus the dominance of Lactobacillus bacteria in the vagina is what confers a "healthy status" to the human vagina because it protects the host from infections.

As you state in your text, the bovine vagina is not an environment dominated by Lactobacillus, and studies fail to identify a single genus of bacteria to be ubiquitously present and dominant in the vagina of all cows (as it commonly is in humans). In addition, you state that great diversity prior to calving was correlated with decreased incidence of reproductive disease. In that case, it seems like maintaining a diverse environment, rather than an environment dominated by a single genus of bacteria, like it is the human vagina, to be the factor in the cow preventing a single strain of pathogenic bacteria to thrive and cause disease. You can then discuss which genera of bacteria have been linked with reproductive diseases and how the less diverse environment characterizes an environment more susceptible to disease. I believe that if you take this approach, it will drive better your point, and facilitate the reader to understand what is known about the vaginal microbiome in the bovine

Line 72: I would rephrase to "reproductive life" as "reproductive cycle" seems like you are reffering to the estrous cycle

Lines 76-78: This sentence is not clear for a reader that is not familiar with the microbiome field. a decreased bacterial richness indicates reduced relative abundance of all bacteria, or that few bacteria are thriving in relation to other bacteria?

Lines 78-79: Maybe rephrase to "such decreased bacterial richness was characterized by great relative abundance of bacteria belonging to the genera Histophilus and Mogibacterium at calving and were associated to later development of endometritis.

Line 84: I suggest using "decreased incidence of reproductive diseases" instead of "healthy"

Dairy Cattle Uterine Microbiome

I like the way you organized this section. 

You first address the aspects of the uterine microbiome that are related to greater incidence of disease, making the relation: Microbiome -> Disease.

Later in the session, you discuss what aspects of the microbiome are known to influence fertility, making the relation: Microbiome -> Fertility

Placental Microbiome

Lines 170-171: Were they similar in relative abundance of the different phyla, or were they similar in bacterial richness? In the previous sentence you state there was a low abundance (richness) at the placentomes near the chorioallantoic membrane. Are you still talking about richness? it is not clear

Semen Microbiome

Line 220: What is considered exotic breeds?

Beef Cattle Vaginal Microbiome

I believe that comparing to the human vaginal microbiome in this section would be beneficial again.

You discuss that the reproductive hormones might be shifting the bacterial communities. In humans, Estradiol will cause accumulation of glycogen in the vaginal epithelial cells, while Progesterone will cause cytolisis of epithelial cells, making glycogen bioavailable. Lactobacillus bacteria will then use the available glycogen as substrate to grow and produce lactic acid. It is unknown as to how reproductive hormones can be shifting the bacterial communities in the vagina of the cow, but presenting such dynamics of the human vaginal microbiome would allow room to speculate that something similar might be happening in the cow, and would even allow speculation on how the hormones used in estrus synch protocols could be influencing such microbiomes

Line 254: What do you define as a healthy reproductive tract here? Are you talking about disease-free, or about a fertile reproductive tract?

In the study from Ault, they found differences in beta diversity between pregnant and non pregnant cows on day -21 (prior to protocol). One could argue that the naturally more diverse microbiome seen in cows that later became pregnant to be a more healthy microbiome

Line 264: in the days leading up to what?

Line 275: How about the effects of Estradiol? I know that this study has an odd result as far as the most dominant phyla of bacteria found disagrees with the commonly found in other studies, but they do analyze the vaginal microbiome under different estradiol concentrations.

Swine Colostrum and Milk Microbiome

Lines 384-389: If you wish to add to this, there are several studies done with germ-free animals that enlighten the role of bacteria in colonizing the intestine to produce SCFA such as butyrate, that is essential for immune activation and several other regular functions that would otherwise be compromised in the absence of bacteria

doi: 10.1038/nature12721 

Sheep and Goat Uterine and Vaginal Microbiome

Line 461: I suggest rephrasing to "belongs to the Pasteurellaceae family, which is a known livestock..."

Equine Vaginal and Uterine Microbiome

Lines 488-492: Does the stage of the estrous cycle alter the communities or is the profile of communities stable? This is confusing

Lines 505-506: Is this correct?  "endometriotendometriotequids"

Reviewer 2 Report

I think it is very well described.

Please check the following points.

1. lines 256 and 257: Is [Pedobcter] spelled correctly?

2. line 423 [78].[82]: Please correct.

3. Reference numbers 32 and 33 are missing in the text.

Reviewer 3 Report

The manuscript is well-written and summarized an interesting and important subject. However, I suggest the authors change the organization of the manuscript: first ruminants (bovine, small ruminants), after non-ruminants (i.e., swine), keeping equines in the end.

Regarding the separation between dairy and beef, similarities/divergences between their microbiomes should be emphasized. I'm not sure about the need for two distinct chapters to describe their microbiomes.

The authors summarized the pathogenic bacteria in reproduction for domestic livestock, but It would be interesting to see a summary of the usual (non-pathogenic) communities also. 

Throughout the manuscript, authors referred to the most abundant phyla or genera, but without references to the actual percentages. Please complement the text with the respective numbers or introduce complementary tables.

Add some discussion regarding the influence of pathogens causing clinical mastitis on the reproductive variables of dairy cows. See for instances:

- Dalanezi FM, Joaquim SF, Guimarães FF, Guerra ST, Lopes BC, Schmidt EMS, Cerri RLA, Langoni H. Influence of pathogens causing clinical mastitis on reproductive variables of dairy cows. J Dairy Sci. 2020 Apr;103(4):3648-3655. doi: 10.3168/jds.2019-16841. Epub 2020 Feb 20. PMID: 32089296.

Didn't you find any references to brucella in the bibliography? It is known to be related to abortion in sheep.  

There are some papers about fetal lamb/kid microbiomes not included in this review:

Bi Y, Tu Y, Zhang N, Wang S, Zhang F, Suen G, Shao D, Li S, Diao Q. Multiomics analysis reveals the presence of a microbiome in the gut of fetal lambs. Gut. 2021 May;70(5):853-864. doi: 10.1136/gutjnl-2020-320951. Epub 2021 Feb 15. PMID: 33589511; PMCID: PMC8040156.

Zou X, Liu G, Meng F, Hong L, Li Y, Lian Z, Yang Z, Luo C, Liu D. Exploring the Rumen and Cecum Microbial Community from Fetus to Adulthood in Goat. Animals (Basel). 2020 Sep 11;10(9):1639. doi: 10.3390/ani10091639. PMID: 32932976; PMCID: PMC7552217.

There are several studies about milk microbiomes in small ruminants, e.g.:

- Esteban-Blanco C, Gutiérrez-Gil B, Puente-Sánchez F, Marina H, Tamames J, Acedo A, Arranz JJ. Microbiota characterization of sheep milk and its association with somatic cell count using 16s rRNA gene sequencing. J Anim Breed Genet. 2020 Jan;137(1):73-83. doi: 10.1111/jbg.12446. Epub 2019 Oct 10. PMID: 31602717.

- Esteban-Blanco C, Gutiérrez-Gil B, Marina H, Pelayo R, Suárez-Vega A, Acedo A, Arranz JJ. The Milk Microbiota of the Spanish Churra Sheep Breed: New Insights into the Complexity of the Milk Microbiome of Dairy Species. Animals (Basel). 2020 Aug 20;10(9):1463. doi: 10.3390/ani10091463. PMID: 32825408; PMCID: PMC7552695.

-Toquet M, Gómez-Martín Á, Bataller E. Review of the bacterial composition of healthy milk, mastitis milk and colostrum in small ruminants. Res Vet Sci. 2021 Nov;140:1-5. doi: 10.1016/j.rvsc.2021.07.022. Epub 2021 Jul 27. PMID: 34358776.

- Novac CȘ, Nadăș GC, Matei IA, Bouari CM, Kalmár Z, Crăciun S, FiÈ› NI, Dan SD, Andrei S. Milk Pathogens in Correlation with Inflammatory, Oxidative and Nitrosative Stress Markers in Goat Subclinical Mastitis. Animals (Basel). 2022 Nov 23;12(23):3245. doi: 10.3390/ani12233245. PMID: 36496766; PMCID: PMC9740090.

- Polveiro RC, Vidigal PMP, de Oliveira Mendes TA, Yamatogi RS, da Silva LS, Fujikura JM, Da Costa MM, Moreira MAS. Distinguishing the milk microbiota of healthy goats and goats diagnosed with subclinical mastitis, clinical mastitis, and gangrenous mastitis. Front Microbiol. 2022 Aug 25;13:918706. doi: 10.3389/fmicb.2022.918706. PMID: 36090116; PMCID: PMC9453028.

Taking into account its relevance to animal health, namely mastitis prevalence, I recommend authors add a section about this subject. Additionally, the authors could emphasize the differences between small ruminants and cows. Notice for instances: 

- Zadoks RN, Tassi R, Martin E, Holopainen J, McCallum S, Gibbons J, Ballingall KT. Comparison of bacteriological culture and PCR for detection of bacteria in ovine milk--sheep are not small cows. J Dairy Sci. 2014 Oct;97(10):6326-33. doi: 10.3168/jds.2014-8351. Epub 2014 Aug 6. PMID: 25108858.

Round 2

Reviewer 3 Report

I thank the authors for the effort to accommodate my previous suggestions.

I have still one concern as I notice three putative self-publications:

Smith, M.S.; Hickman-Brown, K.J.; McAnally, B.E.; Oliveira Filho, R.V.; de Melo G.D.; Pohler, K.G.; Poole, R. K. Reproductive microbiome and cytokines of postpartum beef cows in relation to fertility. Department of Animal Science, Texas A&M University, College Station, TX, USA. 2022, Manuscript in preparation.

Hickman-Brown, K.J.; Smith, M.S.; McAnally, B.E.; Cain, J.W.; Seo, H.; Bazer, F.W.; Johnson, G.A.; Wiegert, J.G.; Poole, R.K. Department of Animal Science, Texas A&M University, College Station, TX, USA. 2022, Manuscript in preparation.

McAnally, B.E.; Smith, M.S.; Wiegert, J.G.; Palanisamy, V.; Dass, S.C.; Poole, R.K. Characterization of boar semen microbiome and its association with sperm quality parameters. Department of Animal Science, Texas A&M University, College Station, TX, USA. 2022, Manuscript in preparation.

Have those works been published already?

If not, my advice is to remove those references.

Author Response

See attached word file. Thank you!
